# Discovery of the Environmental Factors Affecting Urban Dwellers’ Mental Health: A Data-Driven Approach

**DOI:** 10.3390/ijerph17218167

**Published:** 2020-11-05

**Authors:** Chao Wu, Pei Zheng, Xinyuan Xu, Shuhan Chen, Nasi Wang, Simon Hu

**Affiliations:** 1School of Public Affairs, Zhejiang University, Zhejiang 310027, China; zhengpei233@zju.edu.cn (P.Z.); 3180103031@zju.edu.cn (S.C.); 3180102845@zju.edu.cn (N.W.); 2School of Management, Zhejiang University, Zhejiang 310027, China; 3170101653@zju.edu.cn; 3School of Civil and Environmental Engineering, ZJU-UIUC Institute, Zhejiang University, Haining 314400, China; simonhu@zju.edu.cn

**Keywords:** urban data, city environment, mental health, data-driven approach, model comparison

## Abstract

Mental health is the foundation of health and happiness as well as the basis for an individual’s meaningful life. The environmental and social health of a city can measure the mental state of people living in a certain areas, and exploring urban dwellers’ mental states is an important factor in understanding and better managing cities. New dynamic and granular urban data provide us with a way to determine the environmental factors that affect the mental states of urban dwellers. The characteristics of the maximal information coefficient can identify the linear and nonlinear relationships so that we can fully identify the physical and social environmental factors that affect urban dwellers’ mental states and further test these relationships through linear and nonlinear modeling. Taking the Greater London as an example, we used data from the London Datastore to discover the environmental factors that had the highest correlation with urban mental health from 2015 to 2017 and to prove that they had a high nonlinear correlation through neural network modeling. This paper aimed to use a data-driven approach to find environmental factors that had not yet received enough attention and to provide a starting point for research by establishing hypotheses for further exploration of the impact of environmental factors on mental health.

## 1. Introduction

Mental health is an inseparable aspect of modern human health, according to the World Health Organization (WHO), which includes “subjective well-being, perceived self-efficacy, autonomy, competence, inter-generational dependence, and self-actualization of one’s intellectual and emotional potential, among others” [1].

Mental health has been widely studied in many fields, and genetic factors have become a focus of research because of their causal relationship with mental illness in pathology [2,3,4,5]. However, mental health problems are not only caused by mental illness. One fact that cannot be ignored is the interaction of multiple factors, among which environmental components play an important role that lead to mental health issues.

People are constantly exposed to different types of environments. Particularly, these environments can be roughly divided into physical and social environments. These settings can affect an individual’s mental health in a variety of ways, and the most studied fields are the spatial environments related to the neighborhood [6,7], but other environmental factors also have a non-negligible impact. Particulate pollution in the physical environment can cause inflammation of the central nervous system, thereby increasing the chances of mental problems [8,9]; smoking not only does not relieve stress but also increases the risk of mental illness [10,11]. Green spaces which refers to vegetation (trees, grass, forests, parks, etc.) can also have a large impact on mental health [12,13], as noise, crowding, and community escape exits in this physical space can affect mental health [14]. Social environmental factors refer to socioeconomic, racial and ethnic, and relational conditions that may influence a person’s ability to cope with stress. The newly acquired social environment is more likely to cause changes in one’s mental state, such as a decrease or increase in social participation and integration in school, the workplace, and the community [15]. These occurrences are due to the fact of poverty [16,17], the working environment, etc. [18,19]. It is difficult to explain the impact of a certain type of environment on mental health or to explain the actual impact of the complex interaction of factors on mental health. However, access to environmental data is indeed available to us, as sensors throughout the city provide a large amount of physical environmental data, and social statistics provide us with social environmental data. All of these data help us further explore the effect of environmental factors on mental health.

### 1.1. Urban Mental Health

As far as an entire city is concerned, the mental effects it imparts on individuals are important characteristics. Urban centers are not only a place where people live but also an object they shape and reconstruct. Researchers have introduced computational approaches to urban culture that enable people to share their interpretation of the urban environment [20,21]. Therefore, understanding and analyzing the health effects of living in an urban center is worthy of discussion. As the urbanization process continues to advance, an increasing number of people are pouring into cities, exacerbating the impact of the urban environment on people’s mental health. The increasing impact of the urban environment on people’s mental health also requires us to better identify the environmental factors that affect people. However, we still lack an overall understanding of these urban characteristics, and there are insufficient studies on how the environment affects mental health. Moreover, the United Nations believes that the current trend in urbanization is highly unsustainable and must be addressed [22]. Unfortunately, there is currently no official guide to ensure a healthier and more organized form of urbanization for the world’s cities [23,24]. Identifying which factors are associated with mental health issues of individuals living in urban centers is a key element of the development of urbanization. The process of urbanization has brought with it many conveniences, which itself helps us to better record the relevant environmental factors that affect urban dwellers’ mental health. The research of smart cities provides a relevant reference for this article to explore the effects of the city on mental health. 

### 1.2. Urban Data Analysis

As early as 2008, IBM put forward the concept of the “Smart Earth: Next Generation Leadership Agenda” [25]. Smart cities require the use of various information technologies and innovative concepts to improve resource utilization efficiency, optimization of city management and services, and achievement of corresponding goals. The maturity of information and communication technology enables artificial intelligence to explore the impact of environmental factors on cities centers such as urban computing [26]. The mature artificial intelligence algorithms used in urban computing provides us with a reference for exploring the impact of the urban environment on mental health from another perspective. It can be seen that continued excavating the advantages of data analysis is an inexorable trend.

This study aimed to comprehensively analyze the correlation between environmental factors and urban mental health effects. However, the traditional method showed a powerless state in the face of a large amount of dynamic data. The prediction of an urban mental state effect involves a large amount of temporal and spatial data. To obtain a good prediction effect, we first needed to control our input. With sufficient data and time, it is fine to use all the input features, including those irrelevant features, to approximate the underlying function between the input and the output. However, in practice, two problems may be evoked by the irrelevant features involved in the learning process. These are: the irrelevant input features will induce greater computational cost and the irrelevant input features may lead to overfitting.

The feature selection problem has been studied by the statistics and machine learning communities for many years. In traditional statistical feature selection, it is mainly divided into filter methods and wrapped around methods [27]. Filter methods are generally used as a preprocessing step. The selection of features is independent of any machine learning algorithms. Instead, features are selected based on their scores in various statistical tests for their correlation with the outcome variable. The correlation is a subjective term here. In wrapper methods, we tried to use a subset of features and trained a model using them. Based on the inferences that we drew from the previous model, we decided to add or remove features from the subset. The problem was essentially reduced to a search problem. These methods are usually computationally very expensive. 

In multi-feature selection, traditional feature selection methods are difficult to apply. Due to the special nature of urban mental and mental health state analysis, we could not predict a linear or nonlinear relationship between variables in advance, nor could we determine which of the many environmental factors would have a greater contribution. This forced us to use a machine learning algorithm with good generality and low computational complexity for feature selection.

The linear model had a strong explanatory power to the variables to better understand the relationship among the variables for interference, which is mostly used in the field of humanities and social sciences. Although mental state was used as a psychological indicator, the mental state associated to urban areas was affected by multiple factors. Therefore, it was difficult to adapt to linear models that were poor interpretation of the interactions between variables and weakly fit nonlinear relationships.

Although many methods and measures have been applied to linear models to improve their application capabilities, such as local weighting or regularization [28], etc., as a complicated issue, environmental factors, the number of which can be quite high, cannot just use a linear model to explain their impact on urban mental health. Therefore, we used machine learning methods to study the influencing factors of urban centers’ effect on mental health under the multi-characteristics of big data, hoping to determine urban environmental factors worth studying. Through feature selection, mining urban environmental factors, and then confirming the high correlation between corresponding environmental factors and mental health by prediction results.

### 1.3. Determining the Environmental Factors Affecting Urban Dwellers’ Mental Health

The relationship between variables can be statistically divided into two categories: deterministic function relationships and statistical relationships [29]. When we explore the relationship between environmental factors and urban effects on mental states, obviously, we cannot find a simple deterministic functional relationship. The combined effect of multiple factors is presented through statistical relationships, but the impact of different environmental factors on mental states is something we need to judge before making predictions. In linear models, because of the strong interpretation between variables, the method of feature selection also tends to judge linearly related measured indicators. However, in the face of high-dimensional data, it is difficult to use the prior knowledge of the researcher to select features and then use simple metrics to verify. Therefore, we chose the maximal information coefficient (MIC) [30] as the measurement index. MIC is used to measure the degree of association between two variables including linear and nonlinear relationships, ranging from 0–1. Compared with current commonly used coefficients for measuring correlation, such as Pearson’s correlation coefficient [31], distance correlation [32], etc., MIC has a superior way to calculate data relevance, as it is a generality [30]. “Generality” means that when the sample size is large enough or including the majority of the information of the sample, it can capture a variety of interesting associations, and it is not limited to specific function types such as linear functions, exponential functions, or the periodic function.

In addition to the linear model mentioned in the previous section, support vector machines are suitable for multi-factor modeling analysis due to the fact of their ability to model nonlinear relationships, good generalization performance, and the ability to achieve global unique optimization [33]. Artificial neural networks deal with complex problems by simulating the interaction process of neurons [34]. Therefore, by modeling three different levels of models, we could further test whether the selected environmental features reflected changes in urban dweller’s mental health. When the predicted values of the three models were far from the true values, it showed that we needed to adjust the selected features accordingly; when the model achieved good prediction results, it further confirmed the reliability of the environmental factors we selected.

In the study, using models to explore the relationship between the environment and mental health, the purpose of the research was to analyze the factors with significant influence to determine causal relationships. With this research, certain factors were often classified, and cross-sectional studies or multi-level analyses were used to confirm the true effect of research hypotheses in a certain scenario. Taking neighborhood environmental analysis as an example, research on the impact of neighborhood environmental impact on population health has existed for a long time and has shown exponential growth in the initial period to the amount of studies [7]. Measuring the physical and social environment to explore the impact on the community’s mental health is important. Depression has been a mental health outcome most commonly studied concerning neighborhood characteristics.

Measuring mental health has often been regarded as more difficult than measuring other types of health. This is due in part to the limited availability of objective biological tests and variable diagnostic guideline offered by psychiatry, alongside the intercultural differences in the mental health experiences and complex social and psychological confounders. However, it is possible—and desirable—to measure mental health outcomes in environmental research. This is how the mental health impact of urban planning and design can be demonstrated and understood.

With the emergence of a large number of cities and collection of personal data, there are multiple methods to measure mental health. Support vector regression is the most frequently used prediction method. Linear regression also appears in some papers. In addition, with the promotion of machine learning technology, deep learning technology has become a trend, including straightforward deep neural networks and recurrent neural networks. Furthermore, research on measuring mental health through the environment is also emerging such as measuring mental health through the built environment [35] and predicting the impact of economic factors on mental health, etc. [36]. Therefore, we believe that after the environmental factors have been screened out, predictions through the above three models can reflect their correlation.

The promise of big data has made some in the scientific community lazily replace causality with correlation. In the urban mental state, the same correlation obtained through modeling cannot directly explain the role of environmental factors, especially in the interweaving of various environmental factors. However, the modeling of big data does give us the determination of relevance, which is equally important for discovering unknown causality. Therefore, this research focuses on establishing a starting point for future research, obtaining a series of environmental features through feature selection, and determining the credibility of the features based on the comparison between the predictive modeling results and the real situation.

Smarter London Together [37] is a plan initiated by the Greater London Government to meet the needs and challenges of urban residents, workers, and travelers in the digital age. The London Datastore is an internationally recognized open data resource with over 700 data sets that provided data support for our collection of London environmental data. In addition, London’s technology and data on the environment, traffic, safety, etc., showed the data of the entire Greater London area, allowing us to analyze the mental health status of the Greater London area using borough as a unit, and obtain key factors affecting the overall environment of Greater London.

## 2. Materials and Methods

In the case study, Figure 1 showed the flow diagram of the experiment, we collected environmental factors and personal well-being (i.e., happiness) data from questionnaires in the London Datastore including 3 years of environmental data and urban mental health data from 32 boroughs. We screened the top ten features with the highest influence among London environmental factors through the maximal information coefficient (MIC) algorithm, after which three models were built and then environmental factors were analyzed for future hypothesis building.

### 2.1. Study Case and Data Set

In 2006, The Guardian launched the “free our data” campaign, and in 2010 the Open Government Licence and Data.gov.uk websites were launched, a testament to the success of the campaign. 

A dynamic city provides us with more data, and a data-driven approach keeps us from being limited to a specific data type, and structured data, semi-structured data, and unstructured data can all be included in the scope of the study; thus, the data sources were greatly expanded. The government, as a producer and collector of public data, can accurately calculate relevant data in the economic, social, and environmental fields, and the open movement of government information allows us to use these data conveniently.

The London Datastore, which is affiliated with the Greater London Authority (GLA), is a free and open data-sharing portal where anyone can access data related to the capital. The London database provides 1375 types of indicators including both the physical and social environments. The specific indicators are shown in Table 1.

Choosing the appropriate environmental data the modeling analysis with such a large range to explore the relationship between the environment and urban happiness is a difficult task. As mentioned earlier, it is largely limited by prior experience. In this experiment, without directly mentioning environmental factors, the data were initially collected based on 32 boroughs which were the experimental subjects. To ensure the integrity of the data and that there was sufficient data for analysis, we selected environmental data from 32 boroughs over three years from 2015 to 2017. The specific number of fields was 67 in 2015, 41 in 2016, and 34 in 2017 (See the Appendix A for the initial data sets).

The mental state of residents is closely related to environmental factors, and residents living in different regions have different levels of mental health (happiness), the London Datastore provided us with public data on the personal well-being (i.e., happiness) of the London Borough, and through this data, we obtained an accurate mental state individuals living in London. Personal well-being by borough was estimated from the annual population survey (APS) of the well-being data set. The respondents were asked the following questions, for which responses in the range zero to ten were given, where zero was “not at all” and ten was “completely”. 

Anxiety: “Overall, how anxious did you feel yesterday?”Happiness: “Overall, how happy did you feel yesterday?”Life Satisfaction: “Overall, how satisfied are you with your life nowadays?”Worthwhile: “Overall, to what extent do you feel the things you do in your life are worthwhile?”

The environmental data and the personal well-being (i.e., happiness) data of the London Borough were used as the independent variables and dependent variables of the case study in this paper for feature selection and predictive modeling.

Data sets were collected and pre-processed (missing data interpolation, cleaning, and alignment in temporal scale) via Python scripts. All data were then imported into a MongoDB cluster, which is a NoSQL database, to support flexible and scalable data query and analysis for large-scale data sets. These indicators were then extracted from running queries on a MongoDB database. Such data collection and management infrastructure were then applied to other data-driven urban analysis and modeling systems. 

### 2.2. Feature Selection

Research on the environmental factors affecting mental health has a long history, and with the deepening of research, more factors that have not received enough attention in the past have been unearthed. The term “environment” encompasses three major areas [38,39]: the built environment, the natural environment, and the social environment. These three factors affect people in various aspects, so the impact of a certain type of environment on mental health and the interactions among them are numerous. The built environment includes buildings, spaces, and products that people create or modify. Houses, roads, traffic noise, transportation systems, buildings, public places, and urban green spaces all affect people’s mental health [35,40,41,42]. The natural environment includes air, water, and chemical element pollution. For example, the impact of air pollution on mental health is gradually gaining attention [43,44]. The social environment includes social relations, culture, economy, etc. The research on the influence of the social environment on mental health is also becoming more important as the interaction among people deepens and social problems become more prominent [36,45,46].

From the above research and analysis, we can find that the discovery of environmental factors affecting mental health is far from over. How to analyze these indicators that can affect or even predict changes in mental health is worth discussing. The emergence of big data related to urban centers has concretized various environments in the city and, at the same time, displayed urban life in a more granular form, allowing us to analyze us to analyze its unknown factors. 

Through feature selection, we can find environmental factors that are more relevant to mental health in a larger range, and in this way provide a starting point for establishing hypotheses for future research.

In the case of this article, a two-dimensional feature matrix of environmental factors and personal well-being (i.e., happiness) data by borough was constructed. According to the data from the London Datastore, the personal well-being (i.e., happiness) data in 32 boroughs from 2015–2017 including 66 types of environmental factors were gridded, since each grid’s matrix should be calculated. However, when the amount of data was too large, the calculation would become very cumbersome. We used an approximate calculation method; that is, the vertical axis of each resolution was equally spaced, after calculating the MIC, exchanging the horizontal and vertical coordinates, and recalculating the MIC, then taking the maximum value as the final MIC value. In terms of calculation, Python minepy provides a library for the maximal information-based nonparametric exploration (MIC and MINE family) giving an efficient MIC analysis approach.

### 2.3. Models

The feature selection method uses machine learning to help get rid of the limitations of traditional feature selection, since we did not establish pre-model assumptions during feature selection, and because MIC is a generality, we could not only focus on linear models when modeling. Linear models can make a prediction, but the accuracy needs to be promoted, more importantly, the scope of the application of the linear model itself is very limited. On the contrary, the use of nonlinear models, which include support vector regression and neural networks can provide an alternative method of prediction. Exploring whether environmental variables can predict mental health at the linear and nonlinear level was the purpose of this article’s modeling. It helped us to further determine the reliability of the corresponding environmental variables obtained through data-driven methods.

The largest difference between a data-driven approach and the traditional approach to modeling is demonstrated through a thinking perspective. When we explored the relationship between dynamic indicators and urban effects on mental state in cities, we hoped to discover and accurately describe them, which is often expressed through mathematical models. How to choose a mathematical model and determine its parameters is the key to this research. In traditional research, simple models are popular due to the fact of their strong explanatory and low-dimensional parameter determination methods. However, it is very difficult to determine the model of a complex city. We could only make a preliminary idea by combining the models or using a typical neural network method; after that, a large amount of data was used to fit and find the parameters that were closest to the real model. This data-driven approach fits well with the research background in the era of big data and has sufficient basic models to support it. This problem-solving methodology can be introduced with general data-driven thinking and adapted for specific problems. 

Prediction is based on determining the substance of an object’s development and change after corresponding investigation and analysis of the past and present of the predicted object and predicting the future development and change of the object based on the found substance. Forecasting technology is the key to achieving accurate forecasting results. The core of prediction is to predict reoccurring events (prediction rules) through appropriate prediction models and parameters, and the core of prediction technology is to construct the most suitable prediction model for this type of problem through research and analysis. We should realize that the selection of predictive models is an important challenge. No model can solve all problems and can completely fit all situations. Therefore, when predicting urban happiness through environmental factors, especially after the “disenchantment” of prior experience, choosing mainstream and representative models for prediction is the starting point for analyzing this problem. To compare models, we chose the more commonly used and easy to understand mean square error (MSE) for measurement.

#### 2.3.1. Linear Regression

Linear models are widely used in the social sciences. Using a linear model to understand whether the impact of environmental factors on urban dwellers’ mental states is simple and powerful; we also modeled and predicted a linear model to provide a complete analytical perspective. A regression model often relies heavily on the underlying assumptions being satisfied. A regression analysis has been criticized as being misused for these purposes in many cases where the appropriate assumptions cannot be verified to hold. One important factor for such criticism is because a regression model is easier to be criticized than it is to find a method to fit a regression model [47]. However, checking model assumptions should never be an oversighted in regression analysis; therefore, if the environmental variables we selected can obtain good results through a linear model, it just proves that they have a direct and strong relationship with the urban spirit.

Linear model prediction can provide strong support for a causal relationship among variables and can analyze a linear relationship between two variables intuitively and quickly. Linear prediction includes a researchers’ assumptions about the correlation between the dependent variable and the independent variable at the early stage of the hypothesis. It has certain advantages for intuitive causality prediction. When exploring the impact of individual factors on health, such as age, gender, smoking, working environment, air quality, etc., linear models are efficient and excellent. Therefore, whether a linear model can be applied to expose the influence of multiple factors on mental health in a certain urban area was attempted in this article.

#### 2.3.2. Support Vector Regression

To make up for the shortcomings of linear models that are difficult to fit in nonlinear relationships, we first considered support vector regression (SVR) [48], which is an application of support vector machine (SVM) to regression problems. Since the creation of SVM theory by Vapnik in 1995 at the AT&T Bell Laboratories [33], the application of SVM to time series forecasting has shown many breakthroughs and plausible performances. The decisive role of minority support vectors can not only help us grasp key relationships but also provides strong robustness, combined with many successful SVR predictions, which encourage our research using SVR for environmental factors modeling. 

SVR uses the kernel functions technique to transform nonlinear problems into linear problems. The essence is to calculate the distance between two observations. Compared with linear models that easily cause overfitting on high-dimensional data sets, SVR can handle high-dimensional data sets well and has already achieved many mature applications. Through SVR, we can explore the impact of multiple environmental factors on mental health and supplement the deficiencies of the linear models. SVR can model nonlinear problems through several optional kernel functions, and it is highly robust, especially in high-dimensional space. However, there is no universal solution to nonlinear problems. The selection of kernel functions in SVR requires considerable skills, and it is not suitable for large data sets. The kernel function, as a kind of hyperparameter, needs to be specified in advance. When selecting, we used Scikit–Learn’s GridSearchCV to search the kernel function, bringing it into the model, and selected the best-performing parameter. In addition, to reduce the impact of the initial data division results on the performance of the parameters, GridSearchCV uses cross-validation to reduce contingency, and we chose three-fold cross-validation in urban happiness prediction.

#### 2.3.3. Neural Network 

In the nonlinear model, we added a neural network model as a complement to the SVR model. Back propagation neural networks (BPNNs) [49] are the most widely used algorithm models in artificial neural networks. They have high accuracy and strong parallel distributed processing capabilities, distributed storage, learning capabilities, and robustness, and their fault tolerance to noise can fully approximate complex nonlinear relationships. Many neural network models can fit correlation relationship well when the amount of data is sufficient, and this has been applied in various fields. In this study, because of its rigorous derivation process, BPNN was used to help us understand the highly nonlinear mapping characteristics, and the characteristics that were usually applied for supervised learning were also compatible with our experiments and could be adapted to our modeling. The standard back propagation (BP) learning algorithm uses an error function to learn by gradient descent to minimize the mean square error between the actual output value and the expected output value of the network.

The learning process of the BPNN is mainly composed of the forwarding propagation of the input signal and backward propagation of the error. In forward propagation, the input signal is input from the input layer and processed by the hidden layer and then passed to the output layer. If the actual output of the output layer does not match the expected output, then the process goes to the error back propagation process. The process of back propagation consists of modifying the connection weight layer by layer from the output layer to the input layer to reduce the error. The forward propagation of the input signal and the backward propagation of the error are cyclical. The training process for the BPNN is a process of continuously adjusting the connection weights. This process is performed until the output mean square error reaches the required standard.

In the feedforward neural network, since the sigmoid activation function is prone to the phenomenon of “gradient disappearance”, we generally used the rectified linear unit (ReLU) as the activation function of the hidden layer neuron. Compared with the sigmoid function, the main changes were: (1) unilateral inhibition, (2) a relatively wide excitatory boundary, (3) sparse activation. This was close to the working principle of a human neocortex. Another important aspect of neural network training is setting the rate of learning. In this case, the Adam optimizer was used to calculate the adaptive learning rate of each parameter [50]. The method is straightforward to implement, computationally efficient, has little memory requirements, is invariant to the diagonal rescaling of the gradients, and is well suited for problems that are large in terms of data and/or parameters. The adoption of Adam enables the neural network to achieve high-quality training results.

## 3. Results

### 3.1. Feature Selection

Using the MIC algorithm, we analyzed the correlation between environmental factors and urban mental health and obtained a considerable number of findings that exhibited no prior bias or human interference. These results not only further confirm our rationale for this current study but also provide a reference for future correlation analysis.

The ten factors with the highest MIC scores are shown in Table 2, Table 3 and Table 4. The modeling results of these ten factors provide us with a reference for further understanding their relationship with urban mental health, whether it is an intuitive linear relationship or a nonlinear relationship that blends and influences each other, which not only gives us a direct impression but also lays the foundation for the future research.

The 10 features selected by the MIC obtained corresponding results after modeling in the three different ways. Establishing the model, itself, contains the researcher’s concept in forecasting, and different models also reflect different forecasting attitudes. Prediction is a judgment of future situations based on known information and through a strictly logical relationship. Different modeling methods have different concepts, and good modeling results can prove the established hypothesis.

### 3.2. The Comparison of Three Models

When predicting the impact of multiple environmental factors on the mental health of urban dwellers, the modeling of the linear mixed effect model was unsatisfactory and was insufficient to explain the effects of the nonlinear relationships. The mean values of the thirty linear regression experiments from 2015, 2016, and 2017 were 1.36, 1.15, and 0.90, respectively. We recorded the intercept, regression coefficient, and MSE data obtained from each run to provide intuitive results for establishing accurate predictions. The experimental results of the linear regression were not satisfactory. The same environmental factors produced unstable correlation coefficients during the experiment. The influence of some environmental variables on urban happiness was at times positive and at others negative, showing great randomness. The erratic behavior of this factor also showed that the linear model was not suitable for the multi-feature analysis of urban mental health. In other words, in the linear model, the relationship between environmental factors and mental health was very weak, and it was not strong enough to support our exploration of the two.

In the three-year support vector regression experiment from 2015 to 2017, the kernel functions selected by GridSearchCV were all radial basis function (RBF) kernel. The average MSE values of the experiment are shown in Table 5. The overall performance of the SVR was better than the multiple linear regression, and in 2016, the prediction effect through SVR was better than in the other years. To a certain extent, this was determined by the different selection of environmental characteristics. Although we cannot simply determine the specific impact of environmental factors on urban dwellers’ mental health through SVR, we can only determine whether there is a large correlation; however, the small error between the predicted value of SVR and the true value of urban mental health gives us enough confidence that the environmental factors used in the modeling had a potentially nonlinear correlation with urban mental health. Continual exploration of the deep-seated reasons for these factors is the key to any subsequent interpretation of the SVR prediction results.

The greatest feature of neural networks is their ability to simulate the way neurons work and to use neural networks to make connections between inputs and outputs and then generate predictions. The biggest differences between the above model and neural networks is that it did not impose a preconception-based restriction on variables and did not make assumptions about the relationships among variables but rather mined hidden relationships in the data through neural networks. 

Through our experiments, we obtained the average MSEs for the years 2015 to 2017, which were 0.87, 0.96, and 0.78, respectively. From Table 6, we can see that there were certain fluctuations in the MSEs, which were largely due to the changes in the division of the training and test sets. The performances of the same environmental factor index in the 32 boroughs were different, which caused instability among the experimental results in the different training sets. However, in general, the experimental results of the BPNN performed better than the linear regression model and the SVR model.

Predicting via neural networks impacts on the traditional research paradigm of hypothesis verification as it allows analyzing a large amount of available data, and this data-driven paradigm performs a discovery summarization operation. When the amount of data is small, interpretable models, which can provide scientific, rigorous, and accurate predictions, are sought after by researchers. However, with the rapid increase in the volume of data, a large number of new unknown relationships have emerged. Through direct analysis of data, we can mine unrestricted hidden relationships. In exploring the correlation between environmental factors and the mental health of urban dwellers in Greater London, the overall good performance of the BPNN proves that there is a highly nonlinear relationship between the two. However, as is known, the “black box” characteristics of neural networks prevent us from directly judging the causal relationship among variables, and we need to further explore this aspect.

## 4. Discussion

The final stage of urban development is to adhere to a people-oriented concept and create a smart city that makes people’s lives better. Through accurate and objective information technology, flexible and reliable intelligent services can be provided. As mentioned in the Section 1, many factors influence the urban spirit. With the development of urbanization, the urban environment, including the physical and social environments, has an important impact on people’s mental health. Studies on the relationship between the two from multiple angles are also increasingly gaining attention. Even with the development of the social economy, factors that have been ignored in the past are garnering new attention or have produced new changes. This has forced us to re-examine the environmental factors that affect mental health.

The use of urban data to assist in urban development has already played a role in the fields of urban computing and smart cities. Therefore, learning from existing data analysis methods through the collection of big data and screening out environmental factors that have high relevance for negative impacts on mental health is feasible, operable, and meaningful. We used the Greater London area as the scale and analyzed the data of 32 boroughs in Greater London as the case study for this paper to explore the potential environmental factors that affected the mental health of individuals in the Greater London area. 

Since Microsoft published “Fourth Paradigm-Data Intensive Scientific Discoveries” in 2009 [51], a paradigm revolution in the social sciences has been in full swing. However, the application and methods of machine learning have not yet entered the mainstream of social sciences research, and machine learning methods are only being carried out in a few fields. The mining and interpretation of social phenomena often fall into the optimization and selection of algorithms. For example, Super Learner [52] is a response to this phenomenon, but it is more important to grasp the backbone of the data and predictions and then interpret, utilize, and feedback the mined indicators. It is of great significance to dig out the factors affecting urban dweller’s mental health. This can create a prototype platform that will be of interest to policymakers and the private sector for rapidly assessing environmental stressors and their relationships with social and economic strain. It is true that cities have interdisciplinary characteristics (e.g., industry–university–research). Analysis of the dynamic of cities through data alone may not fully reflect the actual situation, but it is important that data-driven analyses of dynamic cities can find interesting and objective conclusions that lay the foundation for subsequent research.

Through the analysis of the data collected from the London Datastore, we obtained environmental factors from the Greater London area from 2015 to 2017 that had a high correlation with mental health. We used linear regression, SVR, and BPNN to predict and test to what extent these factors’ had an influence. Through a three-year analysis of these factors, we found that some factors had a more frequent impact on urban dweller’s mental health, such as fires, the Brexit Referendum, and population data. The impact of these factors, such forest fires, suburban fires, and urban fires, on people’s mental health has also been revealed in related studies, [53,54,55,56]. The mental health issues associated by the Brexit Referendum are gradually being explored and evaluated [57,58,59,60]. The factors related to population data may involve social factors such as race, ethnicity, gender, etc. [61,62,63,64,65]. Relevant research on these unearthed factors are worth in-depth study, but obviously there is a certain gap in the frequently explored factors such as the built environment and the social environment. Using data-driven methods to discover the factors that have not received enough attention, further research is necessary and is increasingly urgent.

While this this study took the Greater London area as a study case, the workflow can also be applied to other case studies with sufficient data. With the improvement of various urban development indexes and requirements, urban public data and data types are gradually enriched. This provides us with an opportunity to use urban data to analyze urban mental health and compare the differences between different regions, neighborhoods, and even countries.

A data-driven feature selection approach can provide us with a relatively good basis for establishing research hypotheses. However, the difference between physical environmental and social environmental factors on mental health may cause data processing problems and thus affect the results. Whether it is possible to classify physical and social environmental factors at multiple levels and discuss the factors with high impact factors separately is worthy of consideration in subsequent research. In general, it is hoped that this research can provide new ideas and reference significance for the analysis of regional mental health problems. 

## 5. Conclusions

This paper provides a new perspective on studying the factors that influence urban dwellers’ mental health. We analyzed the environmental factors affecting the mental health of individuals living in cities and found that with the acceleration of urbanization, the methods brought by the maturity of information and communication technology need to be further applied to urban centers to explore the emerging environmental factors affecting mental health. Some cases of urban computing and smart cities provide us with considerable reference significance, enabling us to analyze where there is sufficient data. Research has found with the refinement of data, and its increasing granularity, it provides us with more opportunities to explore relevant environmental factors and data-driven methods can explore new influencing factors.

Through the analysis of environmental data in the Greater London from 2015 to 2017, we found that social factors, such as fires and the Brexit Referendum, had a greater impact on mental health, which to a certain extent shows that social environmental factors have a more direct and rapid impact on urban mental health. The predictions of linear models, support vector machine models, and neural network models further confirmed that the interaction behind these element correlations were worthy of further exploration, which provides a starting point for subsequent research to establish hypotheses.

In this paper, we do not want to conduct a specific analysis of the environmental factors that affect mental health in the Greater London area but aim to identify highly relevant influencing factors and discover environmental factors that have not received sufficient attention. This kind of data-driven approach can provide new research perspectives and analyze from urban perspectives at different levels, but we should not ignore the importance of data integrity. In future work, we plan to conduct case analyses at different levels and optimize the limitations of feature selection in the case of insufficient data.

## Figures and Tables

**Figure 1 ijerph-17-08167-f001:**
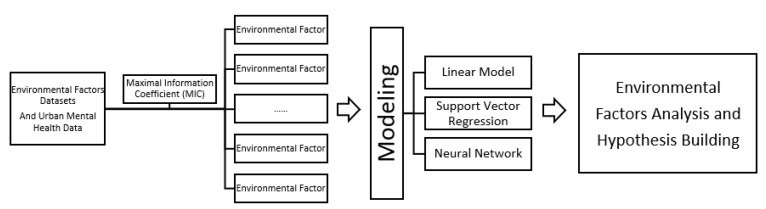
Flow Diagram of Experiment.

**Table 1 ijerph-17-08167-t001:** Indicators in the London Datastore.

Topics	Number of Data Sets
Arts and Culture	27
Business and Economy	108
Championing London	16
Crime and Community Safety	65
Demographics	165
Education	69
Employment and Skills	136
Environment	138
Health	95
Housing	106
Income, Poverty, and Welfare	24
London 2012	5
Planning	104
Sport	26
Transparency	146
Transport	89
Young People	56

**Table 2 ijerph-17-08167-t002:** Ten environmental factors with the highest maximal information coefficient (MIC) values in 2015.

Environmental Factors	MIC Sores
Population by Nationality (Asia)	0.40461737409508647
Home Fire Safety Visits (LFEPA ^1^)	0.3650496777656037
Housing Benefit Caseload	0.3595520749489776
Fires in Other Residential Buildings	0.3462766044158128
The Proportion of the Population Aged 65 and Over	0.32722520966977886
Population by Nationality (European Union)	0.3013517918675558
Dwelling Fires	0.295324652215078
Primary Fires	0.295324652215078
Inland Area (Hectares)	0.29466255649153916
Births	0.28727640812540556

^1^ London Fire and Emergency Planning Authority (LFEPA).

**Table 3 ijerph-17-08167-t003:** Ten environmental factors with the highest MIC values in 2016.

Environmental Factors	MIC Scores
Dwelling Fire Fatalities	0.509123945950592
Dwelling Fires	0.5004404000966433
Fire-Related Fatalities	0.4228406643522162
EU Referendum Results: Leave (%)	0.417606598928714
EU Referendum Results: Remain (%)	0.417606598928714
Population by Nationality (Non-British)	0.38092519966756994
Population by Nationality (European Union)	0.36881105662612673
EU Referendum Results: Remain (Number)	0.33187775400669917
Primary Fires	0.3273617877490797
Home Fire Safety Visits (LFEPA)	0.31287069588326993

**Table 4 ijerph-17-08167-t004:** Ten environmental factors with the highest MIC values in 2017.

Environmental Factors	MIC Scores
PLG ^1^ cars: Private	0.568218701419774
Total PLG	0.568218701419774
Inland Area (Hectares)	0.5487949406953987
Population Density (per hectare)	0.5044054244447898
Exempt Disabled: Cars	0.477367897391955
Household Waste Recycling Rates	0.4074369752008228
PLG: Other Private	0.4065007437523036
Exempt Disabled: Others	0.3112781244591327
GLA ^2^ Population Estimate	0.29071777840020196
Home Fire Safety Visits (LFEPA)	0.27827714327290703

^1^ PLG: Private or Light Goods vehicle (excludes heavy goods, buses and coaches). ^2^ GLA: Greater London Authority.

**Table 5 ijerph-17-08167-t005:** Kernel function and MSE results for the years 2015–2017.

Year	Kernel	MSE
2015	Radial basis function (RBF)	0.93
2016	Radial basis function (RBF)	0.85
2017	Radial basis function (RBF)	0.90

**Table 6 ijerph-17-08167-t006:** Mean square error (MSE) of different models over different years.

Model\Year	2015	2016	2017
LR	1.36	1.15	0.90
SVR	0.93	0.85	0.90
BPNN	0.87	0.96	0.78

LR: Linear Regression; BPNN: Back propagation neural network; SVR: support vector regression.

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
