# Peer review of "Discovery of the Environmental Factors Affecting Urban Dwellers’ Mental Health: A Data-Driven Approach"

_ijerph, 2020, doi:10.3390/ijerph17218167_

Round 1

Reviewer 1 Report

The work develops a new approach of great interest, but it would be necessary to clarify some doubts that would make it more valid. These could be formulated as follows:

1. The predictor variables are defined by the selected databases and although we do not doubt that it is possible to predict the mental state of a population through the data collected in these, I consider that it would be fundamental to establish clear and concise criteria for selecting these, as well as the relevant data for monitoring the state of mental health. In this respect, the article examined is not very clear or concise, and I believe that the author should make an additional effort in this regard. In short, we do not know to what extent the predictor variables are well selected. In this sense, I think it would be good to check to what extent the selected variables are related to the predictors raised in previous epidemiological and prediction studies carried out in the field of mental health. On the other hand, what does this type of analysis add to the work that studies the distribution of mental illnesses by districts, neighbourhoods or postal codes, and then checks the differences in infrastructures and services between these units.
2. The authors themselves are involved in the selection of the predictor variables without any external comparison criteria that would confer greater objectivity.
3. I believe that these comments are important enough to question the validity of the study, so I think that the authors should further clarify these aspects in their discussion and qualify their conclusions.

Author Response

Point 1: The predictor variables are defined by the selected databases and although we do not doubt that it is possible to predict the mental state of a population through the data collected in these, I consider that it would be fundamental to establish clear and concise criteria for selecting these, as well as the relevant data for monitoring the state of mental health. In this respect, the article examined is not very clear or concise, and I believe that the author should make an additional effort in this regard. In short, we do not know to what extent the predictor variables are well selected. In this sense, I think it would be good to check to what extent the selected variables are related to the predictors raised in previous epidemiological and prediction studies carried out in the field of mental health. On the other hand, what does this type of analysis add to the work that studies the distribution of mental illnesses by districts, neighbourhoods or postal codes, and then checks the differences in infrastructures and services between these units.

Point 2: The authors themselves are involved in the selection of the predictor variables without any external comparison criteria that would confer greater objectivity.

Response 1&2:We have reorganized the language. This article aims to find the environmental factors that affect mental health to provide a hypothetical starting point for future work. Therefore, we use data-driven algorithms to find highly relevant environmental factors. We have supplemented the literature on environmental factors used in previous studies to prove that we have sufficient motivation to explore some new environmental factors. The specific supplementary content is located in lines 49-66 and lines 252-277 in the text.

In addition, we only used the overall data of the Greater London area for analysis. The research and analysis of regional impacts need to be strengthened in future work.

Finally, thank you for your valuable review.

Reviewer 2 Report

This is a methodological paper on different machine learning algorithms and methods, and authors proved that they are experts in such data driven approaches, however, they failed to connect in nearly any sense to the original topic: urban mental state.

This paper has its high merits for a community of big data analysts, as it compares different approaches of data analytics and proves how the Neural Network method is best suitable to find relations between large datasets regarding social data. But clearly there was no social scientist in the team, and the paper fails to connect to the scientific discourse of urban mental state, or any social or urban topic it promises to contribute to. In fact there is almost no mention of the urban/social issues by literature review, the datasets used are not presented, the problem addressed is not clearly presented, and most importantly, the results are not presented adequately, and no conclusions on the urban mental state of Greater London are drawn at all.

Some of the problems I found most disturbing:

  • Introduction is fuzzy
  • Discussion is poor, Conclusions are missing
  • Literature review is missing on the topics of urban mental state, big data and smart city research. Even machine learning methodologies lack relevant citations that are highly cited elsewhere
  • Problem introduction for Greater London is missing completely
  • Too extensive description of machine learning algorithms and methodologies in a paper for environmental sciences
  • The initial datasets from greater London are not presented
  • The MSE results are not explained, not clear from tables, not linked to the Environmental factors
  • English language usage is inaccurate, often erratic (like the sentence:"The neurons in each layer are fully connected, but there is no connection between the neurons in the same layer."
  • Tables are not explained, Tables 6-7-8 e.g. are not understandable for an environmental scientist, and probably are superfluous
  • Explanatory figures are missing, existing Figures are not linked to the text 
  • At the end of the paper, the reader does not have any clue about the correlation of environmental factors to happiness, even less about the mental state of Greater London

Author Response

Point:

Introduction is fuzzy

Discussion is poor, Conclusions are missing

Literature review is missing on the topics of urban mental state, big data and smart city research. Even machine learning methodologies lack relevant citations that are highly cited elsewhere

Problem introduction for Greater London is missing completely

Too extensive description of machine learning algorithms and methodologies in a paper for environmental sciences

The initial datasets from greater London are not presented

The MSE results are not explained, not clear from tables, not linked to the Environmental factors

English language usage is inaccurate, often erratic (like the sentence:"The neurons in each layer are fully connected, but there is no connection between the neurons in the same layer."

Tables are not explained, Tables 6-7-8 e.g. are not understandable for an environmental scientist, and probably are superfluous

Explanatory figures are missing, existing Figures are not linked to the text 

At the end of the paper, the reader does not have any clue about the correlation of environmental factors to happiness, even less about the mental state of Greater London

Response:

We revised the framework of the paper as a whole according to your suggestions.
The structure of the introduction has been changed, discussion and conclusion  have been added, relevant documents have been added, and redundant parts have been deleted.
Thank you very much for your patient review. Please refer to the revised manuscript for specific content.

Reviewer 3 Report

Good research.

Just need to clean up the English. Also, I would recommend a stronger discussion and conclusion.

The authors can speculate about the cause of the problems and possible solutions. It's not a problem with discussion. Need to be more than just descriptive and analytical when floating into the social science end.

Also, the stats needs to be explained in simple form for the readership - not everyone knows this, so it's best to clearly and simply describe what it means. Overall, good research.

Just clean up the English so the impact is greater. 

Author Response

Point:

Just need to clean up the English. Also, I would recommend a stronger discussion and conclusion.

The authors can speculate about the cause of the problems and possible solutions. It's not a problem with discussion. Need to be more than just descriptive and analytical when floating into the social science end.

Also, the stats needs to be explained in simple form for the readership - not everyone knows this, so it's best to clearly and simply describe what it means. Overall, good research.

Just clean up the English so the impact is greater. 

Response:

We have increased the literature of the manuscript and revised some conclusions to improve readability, specifically in lines 251-277 and 517-606 in the manuscript. Finally, thank you again for your work and suggestions.

Reviewer 4 Report

The article describes the study of the influence of environmental factors on mental health in cities. Generally speaking, it is a very important topic that is still little discussed in the research, it fits into the profile of the International Journal of Environmental Research and Public Health, but requires a few corrections.

The introduction broadly and interestingly describes the research background related to the progressive urbanization and a number of related factors directly affecting the health of urban residents. The chapter exhaustively describes the research methods used in the observation of these factors, paying attention to the relationship between their effectiveness and the amount of work and costs they require. The chapter could also present the research questions and the structure of the article a little more clearly.

Chapter 2 is far too meager. The scheme of operation (Figure 1) is not clear, the relationships between the various parts of the research are unclear and require a much broader and more precise description than the one presented. The chapter requires a thorough correction.

The above remarks have the effect of an incomprehensible correlation of the successive described elements of the study throughout Chapter 3. This chapter should be supplemented with an additional graphical diagram showing the relationships between the methods described in the subsections. At the moment, the listing of these specifically chosen research methods is unclear and not sufficiently substantiated.

The discussion should be a bit more elaborate. I recommend the following articles:

Ibrahim M., El-Zaarta A., Adams C. (2018). Smart sustainable cities roadmap: Readiness for transformation towards urban sustainability. Sustainable Cities and Society 37 (2018) 530-540

Mori K., Christodoulou A.  (2012) Review of sustainability indices and indicators: Towards a new City Sustainability Index (CSI), Environmental Impact Assessment Review, 32 (2012), pp. 94-106

Uzzell, E. Pol, D. Badenas (2002) Place identification, social cohesion, and enviornmental sustainability, Environ. Behav., 34 (2002), pp. 26-53

The conclusions are too obvious. They require development and reference directly to the methods described in the article.

Author Response

Point:

The article describes the study of the influence of environmental factors on mental health in cities. Generally speaking, it is a very important topic that is still little discussed in the research, it fits into the profile of the International Journal of Environmental Research and Public Health, but requires a few corrections.

The introduction broadly and interestingly describes the research background related to the progressive urbanization and a number of related factors directly affecting the health of urban residents. The chapter exhaustively describes the research methods used in the observation of these factors, paying attention to the relationship between their effectiveness and the amount of work and costs they require. The chapter could also present the research questions and the structure of the article a little more clearly.

Chapter 2 is far too meager. The scheme of operation (Figure 1) is not clear, the relationships between the various parts of the research are unclear and require a much broader and more precise description than the one presented. The chapter requires a thorough correction.

The above remarks have the effect of an incomprehensible correlation of the successive described elements of the study throughout Chapter 3. This chapter should be supplemented with an additional graphical diagram showing the relationships between the methods described in the subsections. At the moment, the listing of these specifically chosen research methods is unclear and not sufficiently substantiated.

The discussion should be a bit more elaborate. I recommend the following articles:

Ibrahim M., El-Zaarta A., Adams C. (2018). Smart sustainable cities roadmap: Readiness for transformation towards urban sustainability. Sustainable Cities and Society 37 (2018) 530-540

Mori K., Christodoulou A.  (2012) Review of sustainability indices and indicators: Towards a new City Sustainability Index (CSI), Environmental Impact Assessment Review, 32 (2012), pp. 94-106

Uzzell, E. Pol, D. Badenas (2002) Place identification, social cohesion, and enviornmental sustainability, Environ. Behav., 34 (2002), pp. 26-53

The conclusions are too obvious. They require development and reference directly to the methods described in the article.

Response:

We have revised the structure of the first section according to your suggestions and added relevant reference; thoroughly revised the content of the second section and reorganized the relationship between different methods. The discussion part has been rewritten and reference have been added.
Specifically in the text 49-121, 251-278 and 517-580 lines.
Finally, thank you very much for your work and suggestions.

Round 2

Reviewer 2 Report

I found that the authors made a considerable improvement to their paper, improving much the understanding and scientific soundness. Now I find the paper meritable to be finished and then published with the following changes:

-  Introduction has been largely improved. I suggest to add sub-chapters to the introduction, and put into these also some parts of the text authors have put in Feature selection and Neural Network chapters. Those parts should go to introduction which explain general concepts, previous studies and the goals of present paper.

Introduction should be divided at least into chapters on

  • without title general goals of the study
  • urban mental health,   
  • smart cities and other studies using big data analysis or urban mental health related data analysis

- Presentation of results have also been improved, but still here is a big critic: It is totally not clear which are the final results. In the conclusions authors state "we found that social factors such as fires, and the Brexit referendum have a greater impact on mental health..." , and MIC scores highlight this, however, the way the three models can correlate to these results, the role of the Model loss diagrams in BPNN, and the whole utility to use BPNN as opposed to the other models is very hard to understand to a reader not experienced in computer sciences, reading this to know something about the utility of the models and the results on urban mental health in London. THIS PART MUST BE MADE MORE UNDERSTANDABLE. Also affecting the conclusions and discussion.

Author Response

Point 1: Introduction has been largely improved. I suggest to add sub-chapters to the introduction, and put into these also some parts of the text authors have put in Feature selection and Neural Network chapters. Those parts should go to introduction which explain general concepts, previous studies and the goals of present paper.

Response 1:We reorganized the structure of the introduction, adjusted some of the content, and added three subsections, specifically in lines 71-220 in the text.

Point 2:Presentation of results have also been improved, but still here is a big critic: It is totally not clear which are the final results. In the conclusions authors state "we found that social factors such as fires, and the Brexit referendum have a greater impact on mental health..." , and MIC scores highlight this, however, the way the three models can correlate to these results, the role of the Model loss diagrams in BPNN, and the whole utility to use BPNN as opposed to the other models is very hard to understand to a reader not experienced in computer sciences, reading this to know something about the utility of the models and the results on urban mental health in London. THIS PART MUST BE MADE MORE UNDERSTANDABLE. Also affecting the conclusions and discussion.

Response 2:We reorganized the results chapter, compared the three modeling results, and described the significance of modeling, specifically in lines 452-526 of the text.

Also, we cleaned our English language and style. At last, thank you very much for your careful work and thoughtful suggestions that have helped improve this paper substantially.

Reviewer 4 Report

The introduction has been extended to include additional research background, which is definitely a positive change.

The illegible diagram (Figure 1) in Chapter 2 has not been corrected. The description of the methods has been made more detailed, the explanations for extending the description of the methodology of the conducted research in chapters 2.2 are particularly positive. and 2.3.3.

In Chapter 3, some schematic would still be desired. Explaining the relationship between the various groups of results.

In Chapter 4,the description of the applications was corrected and made more readable. They are now more specific and relate more directly to the content of the article, previously being too general. Another positive aspect is the expansion of the literature cited list.

Author Response

Point 1:The illegible diagram (Figure 1) in Chapter 2 has not been corrected. The description of the methods has been made more detailed, the explanations for extending the description of the methodology of the conducted research in chapters 2.2 are particularly positive. and 2.3.3.

Response 1:We corrected Figure 1 and made corresponding explanations to reflect the structure of the article.

Point 2:In Chapter 3, some schematic would still be desired. Explaining the relationship between the various groups of results.

Response 2:We reorganized Chapter 3 and compared the results of each model to make it easier to understand, specifically in lines 474-525 in the text.

Also, we cleaned our English language and style. At last, thank you very much for your careful work and thoughtful suggestions that have helped improve this paper substantially.
